# Inhibitory Effects on NO Production and DPPH Radicals and NBT Superoxide Activities of Diarylheptanoid Isolated from Enzymatically Hydrolyzed Ehthanolic Extract of *Alnus sibirica*

**DOI:** 10.3390/molecules24101938

**Published:** 2019-05-20

**Authors:** Hye Soo Wang, Yoon Jeong Hwang, Jun Yin, Min Won Lee

**Affiliations:** Laboratory of Pharmacognosy and Natural Product based Medicine, College of Pharmacy, Chung-Ang University, Seoul 06974, Korea; envi2308@hanmail.net (H.S.W.); g_intention@naver.com (Y.J.H.); yinjun89@naver.com (J.Y.)

**Keywords:** *Alnus sibirica*, oregonin, hirsutanonol, enzymatic hydrolysis, antioxidant, anti-inflammatory

## Abstract

*Alnus sibirica* (AS) is geographically distributed in Korea, Japan, Northeast China, and Russia. Various anti-oxidant, anti-inflammation, anti-atopic dermatitis and anti-cancer biological effects of AS have been reported. Enzymatic hydrolysis decomposes the sugar bond attached to glycoside into aglycone which, generally, has a superior biological activity, compared to glycoside. Enzymatic hydrolysis of the extract (EAS) from AS was processed and the isolated compounds were investigated—hirsutanonol (**1**), hirsutenone (**2**), rubranol (**3**), and muricarpon B (**4**). The structures of these compounds were elucidated, and the biological activities were assessed. The ability of EAS and the compounds (**1**–**4**) to scavenge 2,2-diphenyl-1-picrylhydrazyl (DPPH) radicals and Nitroblue tetrazolium (NBT) superoxide, and to inhibit NO production was evaluated in vitro. EAS showed more potent antioxidant and anti-inflammatory activity than AS. All investigated compounds showed excellent antioxidant and anti-inflammatory activities.

## 1. Introduction

*Alnus sibirica* (AS), belonging to the family *Betulaceae*, is geographically distributed in Korea, Japan, China, and Russia [1]. The Korean Plant Names Index (KPNI), reports that fifteen *Alnus* species are native to Korea. The *Alnus* species are well-known in traditional Korean medicine, such as anti-cancer drugs, cathartics, hemostatics, and skin tonics. AS is used in cough lozenges and as a herbal decoction to treat alcoholism [1]. Previous studies on the chemical constituents of the *Alnus* species have led to the isolation of various diarylheptanoids [2,3,4,5], flavonoids, [6,7], triterpenoids [8,9], and tannins [10,11]. AS is also reported to have anti-oxidant, anti-inflammatory [12], anti-atopic dermatitis [13], anti-adipogenic [14], and cytotoxic [15] properties.

Enzymatic modifications are used to transform compounds isolated from natural sources [16,17,18]. Enzymes are used in various industrial applications where specific catalysts are required. For instance, amylases are used for splitting polysaccharides and proteins in malt, in the brewing industry, and for producing sugars from starch, in food processing industries [11,19]. Aglycones—glycosides in which the sugar molecules have been replaced by hydrogen atoms after enzymatic hydrolysis by intestinal or colonic microflora—are more easily absorbed from the small intestine than glycosides [20,21]. Glycosidation enhances water solubility but reduces chemical reactivity. Therefore, glycosidases play a major role in biological processes and are important in the biological, biomedical, and industrial fields [22].

From our previous study, 17 compounds from fermented AS (FAS) were isolated and evaluated for their antioxidant, anti-inflammatory, and anti-atopic dermatitis activities, in vitro and in vivo, including the quantitative analysis of its components [23,24,25]. The present paper describes the evaluation of antioxidant and anti-inflammatory effects on the enzymatic hydrolysis (EAS). Through this study, diverse but expected chemical and biological changes, such as increased bioavailability or biological activities, were observed, after the glycosides were converted to aglycones.

## 2. Results and Discussion

### 2.1. Enzymatic Hydrolysis

Extracts from *A. sibirica* processed by enzymatic hydrolysis (EAS) was prepared by using Fungamyl Super AX (Novozymes, Bagsvaerd, Denmark). The differences between EAS and AS were observed by thin layer chromatography (TLC). (TLC results data not shown, reaction pathway shows in Figure 1)

### 2.2. Isolation and Structural Identification

The chromatographic isolation of EAS yielded four diarylheptanoids (**1**–**4**). Compound **1** was in the form of an amorphous brown oil. A navy-blue spot was observed after spraying the TLC strip with FeCl_3_ solution. A dark-blue/deep-violet spot was also observed after spraying with 10% H_2_SO_4_ solution and heating. The ^1^H-NMR (600 MHz, DMSO-*d*_6_ + D_2_O) spectrum of **1** showed two aromatic AMX-spin systems indicated by the ortho-meta coupled aromatic signals [δ 6.37–6.36 (2H in total, m, H-6′,6′′)], a meta-ortho coupled aromatic signal [δ 6.59–6.52 (4H in total, m, H-2′, 2′′, 5′, 5′′)], one hydroxyl-bearing methine [δ 3.36–3.32 (1H in total, m, H-5)], and five methylenes [δ 2.67–2.30 (8H in total, m, H-1,2,4,7) and 1.56–1.45 (2H in total, m, H-6)]. (Appendix A) Thus, **1** was identified as hirsutanonol, 1,7-bis-(3,4-dihydroxyphenyl)-5-hydroxyheptane-3-one, after comparison of this spectrum with the reported spectral data for hirsutanonol [26].

Compound **2** was in the form of an amorphous brown oil. A navy-blue spot was observed after spraying the TLC strip with FeCl_3_ solution. A dark-blue/deep-violet spot was also observed after spraying with 10% H_2_SO_4_ solution and heating. The ^1^H-NMR (300 MHz, Acetone-*d*_6_ + D_2_O) spectrum of **2** showed two aromatic AMX-spin systems indicated by ortho-meta coupled aromatic signals [δ 6.94–6.90 (2H in total, m, H-6′, 6′′)], a meta-ortho coupled aromatic signal [δ 7.19–7.11 (4H in total, m, H-2′, 2′′, 5′, 5′′)], an alkene [δ 7.37 (1H, dt, *J* = 13.2, 6.6 Hz, H-5) and 6.57 (1H, d, *J* = 13.2 Hz, H-4)], and four methylenes [δ 3.25–2.89 (8H in total, m, H-1, 2, 6, 7)]. (Appendix A) Thus, **2** was identified as hirsutenone, 1,7-bis-(3,4-dihydroxyphenyl)-4-hepten-3-one, after comparison of this spectrum with the reported spectral data for hirsutenone [26].

Compound **3** was in the form of a dark-yellow oil. A dark-green spot was observed on the TLC strip after spraying with FeCl_3_ solution. A violet spot was also observed after spraying with anisaldehyde-H_2_SO_4_ solution and heating. The ^1^H-NMR (300 MHz, Acetone-*d*_6_ + D_2_O) spectrum of **3** showed two aromatic AMX-spin systems indicated by an ortho-meta-coupled aromatic signal [δ 6.71–6.66 (4H in total, m, H-2′, 2′′, 5′, 5′′), 6.51 (1H, dd, *J* = 6.3, 2.1 Hz, H-6′), and 6.45 (1H, dd, *J* = 6.3, 2.1 Hz, H-6′′)] and six methylenes [δ 3.57–3.51 (1H in total, m, H-5), 2.68–2.42 (4H in total, m, H-1, 7), 1.69–1.42 (8H in total, m, H-2, 3, 4, 6)]. (Appendix A) Thus, compound **3** was identified as rubranol, 1,7-bis-(3,4-dihydroxyphenyl)-5-hydroxyheptane, after comparison of this spectrum with the reported spectral data for rubranol [27].

Compound **4** was in the form of an amorphous brown oil. A navy-blue spot was observed after spraying the TLC strip with FeCl_3_ solution. A dark-green spot was also observed after spraying with 10% H_2_SO_4_ solution and heating. The ^1^H-NMR (600 MHz, Acetone-*d*_6_ + D_2_O) spectrum of **5** showed two aromatic AMX-spin systems indicated by a meta-ortho-coupled aromatic signal [δ 6.68 (1H, d, *J* = 7.2 Hz, H-5′), 6.67 (1H, d, *J* = 7.2, H-5′′), 6.66 (1H, d, *J* = 2.1 Hz, H-2′), 6.64 (1H, d, *J* = 2.1 Hz, H-2′′)], ortho-meta-coupled aromatic signals [δ 6.45 (1H, dd, *J* = 7.2, 2.1 Hz, H-6′), 6.44 (1H, dd, *J* = 7.2, 2.1 Hz, H-6′′)], and six methylenes [2.67–2.61 (4H in total, m, H-1, 2), 2.42 (2H, t, *J* = 7.2 Hz, H-4), 2.40 (2H, t, *J* = 7.2 Hz, H-7), 1.49 (4H, m, H-5, 6)]. The ^13^C-NMR (150 MHz, Acetone-*d*_6_ + D_2_O) spectrum of **4** also showed the presence of an aromatic AMX-spin system indicated by hydroxyl-bearing aromatic carbon signals [δ 144.8, 144.7, 143.1, and 142.9 ppm (C-3′, 3′′, 4′ and 4′′)] in a region downfield from the signals [δ 119.3, 119.2, 115.3, 115.2, 115.1, and 114.0 ppm (C-2′, 2′′, 5′, 5′′, 6′ and 6′′)]. In the upfield region, a ketone group was observed at δ 210.4 (C-3). (Appendix A) Thus, **4** was identified as muricarpon B, 1,7-bis(3,4-dihydroxyphenyl)-3-heptanone, after comparison of this spectrum with the reported spectral data for muricarpon B [28].

### 2.3. Biological Activities

The2,2-diphenyl-1-picrylhydrazyl (DPPH) radical scavenging activities of AS, EAS, and the four compounds were assessed. According to the results (Table 1), EAS (IC_50_ = 16.68 ± 0.37 μg/mL) showed superior DPPH radical scavenging activity to that of AS (IC_50_ = 21.80 ± 0.55 μg/mL). Compounds **1** (IC_50_ = 26.02 ± 0.57 μM), **2** (IC_50_ = 19.39 ± 0.32 μM), **3** (IC_50_ = 23.30 ± 1.00 μM), and **4** (IC_50_ = 38.70 ± 0.71 μM) showed potent DPPH radical scavenging activities, compared with the positive control, l-ascorbic acid.

The Nitroblue tetrazolium (NBT) superoxide scavenging activities of AS, EAS, and the four compounds were assessed. According to the results (Table 2), EAS (IC_50_ = 3.12 ± 0.75 μg/mL) showed a more potent NBT superoxide scavenging activity compared to that of AS (IC_50_ = 4.59 ± 0.68 μg/mL). According to the results (Table 2), compounds **1** (IC_50_ = 19.03 ± 8.79 μM), **2** (IC_50_ = 16.68 ± 6.74 μM), **3** (IC_50_ = 11.62 ± 7.86 μM), and **4** (IC_50_ = 12.65 ± 11.05 μM) showed good NBT superoxide scavenging activities, compared with those of the positive control, allopurinol.

Massive amounts of nitric oxide (NO) produced by the inducible nitric oxide synthase (iNOS) under pathological conditions, for example, inflammatory disease, are potentially harmful, especially when time-spatial regulation of the iNOS expression becomes compromised. During inflammation associated with different pathogens, NO production increases significantly and might become cytotoxic [29]. Moreover, the free radical nature of NO and its high reactivity with oxygen to produce peroxynitrite (ONOO^–^) makes NO a potent pro-oxidant molecule, capable of inducing oxidative damage and being potentially harmful towards cellular targets [30]. Thus, the inhibition of NO production in response to inflammatory stimuli, might be a useful therapeutic strategy in inflammatory disease [31,32]. Inhibitory activities on the NO production of AS, EAS, and the four compounds were assessed. According to the results (Table 3), EAS (IC_50_ = 1.14 ± 0.06 μg/mL) showed better inhibitory effects on NO production, than AS (IC_50_ = 6.26 ± 0.16 μg/mL) and the positive control—NG-methyl-l-arginine acetate salt (L-NMMA) (IC_50_ = 3.53 ± 0.17 μg/mL). According to the results (Table 3), most compounds showed better inhibitory effects on NO production than L-NMMA (IC_50_ = 33.88 ± 27.87 μM). Compounds **2** (IC_50_ = 0.78 ± 0.38 μM) and **4** (IC_50_ = 2.64 ± 2.29 μM) exhibited markedly potent inhibitory effects on NO production.

EAS and AS were measured for their biological activities, and EAS was found to exhibit better activities than AS. Compounds **1**–**4,** isolated from EAS showed potent anti-oxidative and anti-inflammatory effects, especially **2**–**4**. In addition, these compounds showed cytotoxic activity against cancer cell lines [33]; however, in this study, the experiments were carried out in an amount that did not result in cytotoxicity (data not shown). In a previous study that we had carried out [34], **1**–**3** were greatly increased and **4** was newly formed. This seemed to have made a great contribution toward the increased efficacy of EAS.

## 3. Materials and Methods

### 3.1. General Procedure

The stationary phases for column chromatography were carried out using Sephadex LH-20 (10–25 μm, GE Healthcare Bio-Science AB, Uppsala, Sweden) and MCI-gel CHP 20P (75–150 μm, Mitsubishi Chemical, Tokyo, Japan). ODS-B gel (40–60 μm, Daiso, Osaka, Japan) was used as the stationary phase on a medium pressure liquid chromatography (MPLC) system and consisted of an injector (Waters 650E), a pump (TBP5002, Tauto Biotech, Shanghai, China), and a detector (110 UV/VIS detector, Gilson, Middleton, WI, USA). TLC analysis was carried out using precoated silica gel plates (Merck, Darmstadt, Germany) with a mixture of CHCl_3_, CH_3_OH, and H_2_O (80:20:2, volume ratio) as the mobile phase. Spots on the TLC strips were detected by spraying with FeCl_3_ and anisaldehyde-H_2_SO_4_ or 10% H_2_SO_4_, followed by heating. ^1^H-(300 or 600 MHz) and ^13^C-(150 MHz) NMR experiments, as well as 2D-NMR experiments, such as the heteronuclear single quantum coherence (HSQC) and the heteronuclear multiple bond coherence (HMBC) experiments, were performed using VNS (Varian, Palo Alto, CA, USA) and Gemini 2000 spectrometers (Varian), in the research facilities of the Chung-Ang University.

### 3.2. Plant Material

Barks of AS were collected from ‘Kuksabong′, Seoul, Republic of Korea, in January 2015 and authenticated by Professor Lee (College of Pharmacy, Chung-Ang University, Seoul, Korea). The voucher specimen (201501-AS) was placed at the Laboratory of Pharmacognosy and Natural Product-Derived Medicine at the Chung-Ang University.

### 3.3. Enzymatic Hydrolysis

We used Fungamyl Super AX^®^ (Novozymes) for the hydrolysis experiments. The purchased enzyme was mixed with AS extract and distilled water, in the ratio of 3:1:1. The mixture was allowed to react at room temperature for 3 days. After the enzymatic hydrolysis, the enzyme was removed via ethyl acetate fractionation. For this, centrifugation was performed, and the supernatant obtained was mixed with an equal volume of ethyl acetate; this process was repeated thrice. The ethyl acetate layer was then evaporated to obtain EAS.

### 3.4. Extraction and Isolation

The barks of AS (2.8 kg) were extracted with 80% ethanol (30 L) at room temperature. After removing ethanol, the mixture was concentrated to obtain 121 g of AS extract. A part (23.27 g) of this extract was subjected to enzymatic hydrolysis using Fungamyl (to obtain EAS), followed by liquid–liquid partition usingethyl acetate. The rest of the extract was stored in the freezer and the ethyl acetate layer was then subjected to Sephadex LH-20 column chromatography and then eluted with a solvent gradient system of MeOH:H_2_O (from 2:8 to 10:0), yielding eight sub-fractions (EAS-1 to 8). From fraction EAS-2, compound **1** (hirsutanonol, 283 mg) was isolated. When EAS-6 (203 mg) in the ODS gel was subjected to MPLC (flow rate: 5 mL/min) with a gradient solvent system of MeOH:H_2_O (from 0:10 to 10:0), **2** (hirsutenone, 76.9 mg) was obtained. EAS-7 (1.5 g), when subjected to MCI gel open-column chromatography with a solvent gradient system of MeOH:H_2_O (from 6:4 to 10:0) yielded **3** (rubranol, 504 mg) and **4** (muricarpon B, 125.8 mg).

### 3.5. Measurement of DPPH Radical Scavenging Activity

The antioxidant activity was evaluated on the basis of the scavenging activity of the stable DPPH free radical (Sigma, St. Louis, MO, USA). Each sample (20 μL), in anhydrous ethanol, was added to 180 μL of DPPH solution (0.2 mM, dissolved in anhydrous ethanol). After mixing gently and letting it stand for 30 min at 37 °C, in a dark environment, the absorbance was measured at 517 nm, using an enzyme-linked immunosorbent assay (ELISA) reader (TECAN, Salzburg, Austria). The free radical scavenging activity was calculated as the inhibition rate (%) = 100 − (sample O.D./control O.D.) × 100. l-ascorbic acid was used as the positive control.

### 3.6. Measurement of NBT Superoxide Scavenging Activity

A reaction mixture with a final volume of 632 μL/eppendorf tube was prepared with 50 mM phosphate buffer (pH 7.5) containing K-EDTA (1 mM), hypoxanthine (0.6 mM), NBT (0.2 mM) (Sigma, St. Louis, MO, USA), 20 μL of aqueous extract (distilled water for the control), and 20 μL of xanthine oxidase (1.2 U/μL) (Sigma, St. Louis, MO, USA). The xanthine oxidase was added at last. For each sample, a blank reaction was carried out by using distilled water, instead of the extract and xanthine oxidase. NBT reduction was evaluated by determining the absorbance at 612 nm, after incubation at 37 °C for 10 min. Superoxide anion scavenging activities were calculated as 100 − (sample O.D. − blank O.D.)/(control O.D. − blank O.D.) × 100, and were expressed as IC_50_ values, which were defined as the concentrations at which 50% of NBT superoxide anion was scavenged. Allopurinol was used as the positive control.

### 3.7. RAW264.7 Cell Culture

The murine macrophage RAW264.7 cells were purchased from the Korean Cell Line Bank. These cells were grown at 37 °C in a humidified atmosphere (5% CO_2_) in Dulbecco′s Modified Eagle Medium (Sigma, St. Louis, MO, USA), containing 10% fetal bovine serum, 100 IU/mL penicillin G, and 100 mg/mL streptomycin (Gibco BRL, Grand Island, NY, USA). The cells were used in the in vitro experiments, after counting with a hemocytometer.

### 3.8. Measurement of Inhibitory Activity on NO Production

RAW264.7 macrophage cells were cultured in a 96-well plate and incubated for 4 h at 37 °C, in a humidified atmosphere (5% CO_2_). The cells were incubated in a medium containing 10 µg/mL lipopolysaccharide (Sigma) and the test samples. After incubating for an additional 20 h, the NO content was evaluated by the Griess assay. The Griess reagent (0.1% naphthylethylenediamine and 1% sulfanilamide in 5% H_3_PO_4_ solution; Sigma) was added to the supernatant of the cells treated with the test samples. The absorbance at 540 nm, against a standard sodium nitrite curve, was used to determine the NO content. L-NMMA was used as the positive control. NO production inhibitory activity was calculated as inhibition rate (%) = 100 − (sample O.D. − blank O.D.)/(control O.D. − blank O.D.) × 100, and was defined as IC_50_, which was the concentration that could inhibit 50% of NO production.

### 3.9. Statistical Analysis

All data are expressed as the mean ± SD of three replicates. Values were analyzed by one-way analysis of variance (ANOVA) followed by Student–Newman–Keuls test using the Statistical Package for the Social Sciences (SPSS) software pack; a statistical difference was considered to be significant when the *p*-value was less than 0.05. Values bearing different superscripts in the same column are significantly different.

## 4. Conclusions

In order to evaluate the anti-oxidative and anti-inflammatory effects of the EAS and its compounds (**1**–**4**), DPPH radical, NBT superoxide scavenging activities, and inhibitory activity on NO production were evaluated, in vitro. According to the results, the anti-oxidative and anti-inflammatory activities of the EAS were much better than the ethanolic crude extract of AS. Isolated compounds **1**–**4** showed significantly better anti-oxidative and anti-inflammatory activities, compared to their respective positive controls. The contents of **1**–**4** were increased and this appeared to be important in increasing the efficacy of EAS. These results suggest that EAS is a new source for the development of anti-oxidative and anti-inflammatory agents.

## Figures and Tables

**Figure 1 molecules-24-01938-f001:**
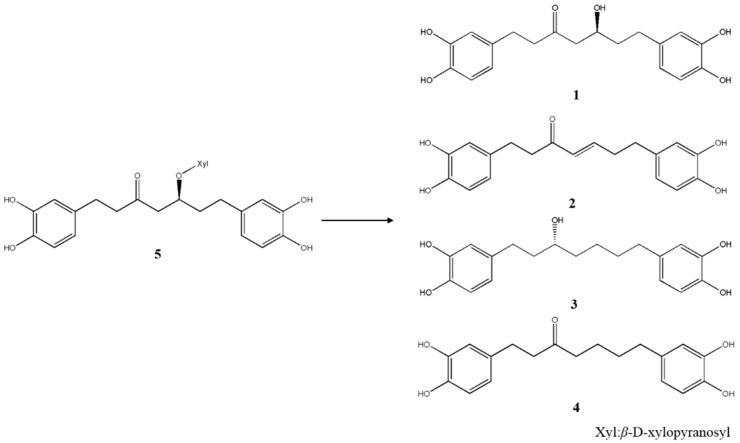
Chemical structure of **1**–**5** and the hydrolysis reaction pathway

**Table 1 molecules-24-01938-t001:** IC_50_ values of *Alnus sibirica* (AS), enzymatic hydrolysis (EAS), and the isolated compounds for the 2,2-diphenyl-1-picrylhydrazyl (DPPH) radical scavenging activity.

Samples	IC_50_ (μg/mL)	Compounds	IC_50_ (μM)
AS	21.80 ± 0.55 ^c^	**1**	26.02 ± 0.57 ^a^
EAS	16.68 ± 0.37 ^b^	**2**	19.39 ± 0.32 ^a^
Ascorbic acid	11.18 ± 0.60 ^a^	**3**	23.30 ± 1.00 ^a^
		**4**	38.70 ± 0.71 ^c^
		Ascorbic acid	44.67 ± 1.75 ^b,c^

Values are presented as mean ± SD (*n* = 3). Values bearing different superscripts (a–c) in same columns are significantly different (p < 0.05).

**Table 2 molecules-24-01938-t002:** IC_50_ values of AS, EAS, and the isolated compounds for the nitroblue tetrazolium (NBT) superoxide scavenging activity.

Samples	IC_50_ (μg/mL)	Compounds	IC_50_ (μM)
AS	4.59 ± 0.68 ^c^	**1**	19.03 ± 8.79 ^b^
EAS	3.12 ± 0.75 ^b^	**2**	16.68 ± 6.74 ^b^
Allopurinol	0.10 ± 0.41 ^a^	**3**	11.62 ± 7.86 ^b^
		**4**	12.65 ± 11.05 ^b^
		Allopurinol	3.92 ± 1.96 ^a^

Values are presented as mean ± SD (*n* = 3). Values bearing different superscripts (a–c) in same columns are significantly different (*p* < 0.05).

**Table 3 molecules-24-01938-t003:** IC_50_ values of AS, EAS, and the isolated compounds for the inhibitory activity on NO production.

Samples	IC_50_ (μg/mL)	Compounds	IC_50_ (μM)
AS	6.26 ± 0.16 ^c^	**1**	17.81 ± 8.63 ^a,b,c^
EAS	1.14 ± 0.06 ^a^	**2**	0.78 ± 0.38 ^a^
L-NMMA	3.53 ± 0.17 ^b^	**3**	5.20 ± 2.61 ^a,b^
		**4**	2.64 ± 2.29 ^a^
		L-NMMA	33.88 ± 27.87 ^b,c^

Values are presented as mean ± SD (*n* = 3). Values bearing different superscripts (a–c) in same columns are significantly different (*p* < 0.05).

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
