# Peer review of "Inhibitory Effects on NO Production and DPPH Radicals and NBT Superoxide Activities of Diarylheptanoid Isolated from Enzymatically Hydrolyzed Ehthanolic Extract of Alnus sibirica"

_molecules, 2019, doi:10.3390/molecules24101938_

Round 1
Reviewer 1 Report
The paper is interesting.
I have some question regarding this paper:
The compounds are cytotoxic? This is important to establish the safety of the product
The conclusion is very short and the Authors should highlight the possible clinical significance of their findings
Author Response
Response to Reviewer 1 Comments
Reviewer 1)
The paper is interesting.
I have some question regarding this paper:
The compounds are cytotoxic? This is important to establish the safety of the product
The conclusion is very short and the Authors should highlight the possible clinical significance of their findings
Answer)
→ The isolated compounds have cytotoxic activity against some cancer cell lines [Choi, S.E.; Kim, K.H.; Kwon, J.H.; Kim, S.B.; Kim, H.W.; Lee, M.W. Cytotoxic Activities of Diarylheptanoids from Alnus japonica. Arch. Pharm. Res. 2008, 31, 1287-1289].
But in this study, experiments on nitric oxide inhibition were carried out in an amount that does not cause cytotoxicity.

Reviewer 2 Report
The presented research has a lot of results, the authors worked very hard in this study, but the quality of the manuscript must be improved.
- The title is not in accordance with the content
- The abstract must be improved, a concise and more clear abstract will be helpful for readers
- The introduction must be improved with data concerning the species
- The Results are presented in section 2, but the Discussions section is missing!
- According to the format of Molecules, Discussion section and Chemicals are demanded
- The extracts preparation is not complete presented (FAS?); what is the solvent “prethanol A”
- The anti-inflammatory activity was not determined, the influence of the extracts on the NO production could be a possible mechanism
- The English in many places of this manuscript is difficult to understand, there are a lot of language mistakes.
Author Response
Response to Reviewer 2 Comments
Reviewer 2)
The presented research has a lot of results, the authors worked very hard in this study, but the quality of the manuscript must be improved.
1) The title is not in accordance with the content
Answer) The title was revised. “Inhibitory effects on NO production and DPPH radicals and NBT superoxide activities of the diarylheptanoid isolated from Enzymatically hydrolysed ehthanolic extract of Alnus sibirica”
2) The abstract must be improved, a concise and more clear abstract will be helpful for readers
Answer) → The abstract was improved as below (Added about Alnus sibirica and enzymatic hydrolysis and delete the content analysis for avoiding confusion.)
[Alnus sibirica (AS) geographically distributes in Korea, Japan, Northeast China and Russia. Various biological effects of AS on anti-oxidant, anti-inflammation, anti-atopic dermatitis ans anti-cancer were reported. Enzymatic hydrolysis decomposed sugar bond attached glycoside into aglycone which is better biological activity than glycoside generally. Enzymatic hydrolysis on the extract (EAS) from Alnus sibirica (AS) was processed and investigated the compounds isolated therefrom: hirsutanonol (1), hirsutenone (2), rubranol (3), and muricarpon B (4). The structures of these compounds were elucidated, and the biological activities were assessed. The ability of EAS and the compounds (1–4) to scavenge DPPH radicals and NBT superoxide and to inhibit NO production was evaluated in vitro. EAS showed more potent antioxidant and anti-inflammatory activity than AS. And all compounds showed excellent antioxidant and anti-inflammatory activities. Compositional analysis of AS, EAS, and the compounds showed that the level of oregonin was considerably decreased, and the levels of 1 and 2 were markedly increased, and 4 was newly formed in EAS. These results indicate that EAS extract has stronger antioxidant and anti-inflammatory effects than AS.]
3) The introduction must be improved with data concerning the species
Answer) → The introduction was improved as below. (Added about origin, chemical constituents of species and how used for medicine)
[Alnus sibirica (AS), belonging to the family Betulaceae, is geographically distributed in Korea, Japan, China, and Russia [1]. According to Korean Plant Names Index (KPNI), it is reported that fifteen Alnus species were native to Korea. The Alnus species are well known for Korean traditional medicine such as anti-cancer drug, cathartic, hemostatic, skin tonics. AS is used as an antipyretic, expectorant, antiphlogistic, and antiasthmatic; it has also been used in cough lozenges and as an herbal decoction to treat alcoholism [1]. Previous studies on the chemical constituents of Alnus species have led to the isolation of variousPhenolic compounds in AS include diarylheptanoids [2-5], flavonoids, [6,7], triterpenoids [8,9], and tannins [10,11]. AS also is reported to have anti-oxidant, anti-inflammatory [12], anti-atopic dermatitis [13], anti-adipogenic [14] and cytotoxic [15] properties.]
4) The Results are presented in section 2, but the Discussions section is missing!
Answer) →Section 2 is “Results and Discussions”, and we added some explanation.
[ EAS and AS were measured their biological activities and EAS exhibited better activities than AS. Compounds 1-4 isolated from EAS showed potent anti-oxidative and anti-inflammatory effects, especially 2-4. And compounds have cytotoxic activity against cancer cell line [33], but in this study, experiments were carried out in an amount that does not cause cytotoxicity (data not shown). In our previous study [34], 1-3 were increased highly and 4 was formed newly. That part seems to have made a great contribution to making EAS more effective.]
5) According to the format of Molecules, Discussion section and Chemicals are demanded
Answer) → Section 2 is “Results and Discussions” and revised Fig. 1 for Chemicals.
Figure 1. Sructures of compounds 1-4Chemical structure of 1-5 and hydrolysis reaction pathway
6) The extracts preparation is not complete presented (FAS?); what is the solvent “prethanol A”
Answer) → The extracts preparation step was described in “Materials and methods” part as below and FAS is fermented Alnus sibirica and detail was described in the reference [23]. Preathanol A is edible ethanol.
[The barks of AS (2.8 kg) were extracted with 80% ethanol (30 L) at room temperature. After removing ethanol, the mix was concentrated to obtain 121 g of AS extract. A part (23.27 g) of this extract was subjected to enzymatic hydrolysis with Fungamyl (to obtain EAS) and followed by liquid-liquid partition usingethyl acetate. The rest of the extract was stored in the freezer and ethyl acetate layer was then subjected to Sephadex LH-20 column chromatography and eluted with a solvent gradient system of MeOH:H2O (from 2:8 to 10:0), yielding 8 sub-fractions (EAS-1 to 8). From fraction EAS-2, compound 1 (hirsutanonol, 283 mg) was isolated. When EAS-6 (203 mg) in ODS gel was subjected to MPLC (flow rate: 5 mL/min) with a gradient solvent system of MeOH:H2O (from 0:10 to 10:0), 2 (hirsutenone, 76.9 mg) was obtained. EAS-7 (1.5 g), subjected to MCI gel open-column chromatography with a solvent gradient system of MeOH:H2O (from 6:4 to 10:0) yielded 3 (rubranol, 504 mg) and 4 (muricarpon B, 125.8 mg).]
7) The anti-inflammatory activity was not determined, the influence of the extracts on the NO production could be a possible mechanism
Answer) → We change the title “Inhibitory effects on NO production and DPPH radicals and NBT superoxide activities of the diarylheptanoid isolated from Enzymatically hydrolysed ehthanolic extract of Alnus sibirica” and described the possibility of the inhibition of NO production in response to inflammatory stimuli might be a therapeutic strategy in inflammatory disorders in the section of “Results and Discussions” as below.
[Massive amounts of nitric oxide (NO) produced by inducible nitric oxide synthase (iNOS) under pathological conditions, for example inflammatory disease, are potentially harmful, expecially when time-spatial regulation of iNOS expression becomes compromised. During inflammation associated with different pathogens, NO production increases significantly and may become cytotoxicity [29]. Moreover, the free radical nature of NO and its high reactivity with oxygen to produce peroxynitrite (ONOO-) makes NO a potent pro-oxydant molecule able to induce oxidative damage, and to be potentially harmful towards cellular targets [30]. Thus the inhibition of NO production in response to inflammatory stimuli might be a useful therapeutic strategy in inflammatory disease [31,32].]
8) The English in many places of this manuscript is difficult to understand, there are a lot of language mistakes.
Answer) → We followed the recommendation to check English pre-edit services from MDPI.

Reviewer 3 Report
Review Comments on the manuscript, “Anti-inflammatory and antioxidant activities of the diarylheptanoid isolated from enzymatic hydrolysed Alnus sibirica by Fungamyl Super AX and comparisons with original plant”
Lines 2-5, “Anti-inflammatory and antioxidant activities of the diarylheptanoid isolated from enzymatic hydrolysed Alnus sibirica by Fungamyl Super AX and comparisons with original plant”
Comment 1. All data and discussion regarding the variation in the content of the isolated diarylheptanoid due to processing using enzymatic hydrolysis must be deleted from this manuscript since this topic has been previously published:
“Wang, H.S., Yin, J., Hwang, I.H., and Lee, M.W., Variation of diarylheptanoid from alnus sibirica fitch. ex turcz. processed enzymatic hydrolysis. Korean Journal of Pharmacognosy, 2018. 49(4): pp. 336-340.”
Inclusion of this data and discussion in this manuscript entails plagiarism.
Furthermore, it is contentious why the authors have not sited this previously published article?
Comment 2. Change the title to, “Anti-inflammatory and antioxidant activities of the diarylheptanoid isolated from enzymatically hydrolysed ethanolic extract of Alnus sibirica”
Line 40, “ In our study, 17 compounds from …..”
Comment 3. These were not part of the current manuscript but rather from the previous reported study, this is confusing to readers and should be changed to … “ From our previous study…”
Line 43, “ on EAS including the compositional analysis of the diarylheptanoid…”
Comment 4. Again, remove this part from this manuscript, this has been reported previously (see comment 1).
Line 51-53, “Figure 1. Stuctures of compounds 1-4”
Comment 5. Remove this figure, it has very strong similarity with the previously published article (see comment 1). It is rather recommended to show the hydrolysis reaction scheme with reaction conditions indicated.
Lines 56. 65, 73, 81, “….1, 2, 3, 4…”
Comment 6. Write instead as, Compound 1, Compound 2 …..”
Line 84, “…[d 6.68 (1H, d, J = 7.2, H-5′)…..”
Comment 7. Be consistent in describing coupling constants, always indicate “Hz” after values.
Line 84, “The 1H-NMR spectrum….”
Comment 8. Similarly to all compounds described, indicate the Hz of NMR machine and d-solvents used.
Comment 9. Furthermore, a complete table of 1H-NMR and 13C-NMR values with pertinent NMR information indicated should be presented as table or supplemental texts.
Line 162, “….under vacuum, the mix was…”
Comment 10. Change to, “…., the mixture was …”
Line 164, “ … fractionation by ethyl acetate; …”
Comment 11. Be specific on how the fraction was obtained, if it was liquid-liquid partition, then it should be changed to, “… followed by liquid-liquid partition using ethyl acetate and…. The ethyl acetate layer was …..and the …. was stored in the freezer.”
Line 164-165, “EAS was then subjected to Sephadex LH-20….”
Comment 12. Specify which EAS solution was used in this section, the aqueous, ethanolic or the ethyl acetate layer?
Line 167-168, “….was subjected to MPLC ….”
Comment 13. Specify the MPLC conditions (detection wavelength, injection volume etc…) and columns (brand, particle size, i.d.) used.
Line 176, “…standing for 30 min…”
Comment 14. Specify incubation parameters (e.g. temperature and light conditions) and detector wavelength used.
Line 184, “… a blank reaction was carried out…”
Comment 15. Specify the composition of the blank reactants.
Line 224, “… the EAS were much better than those of AS.”
Comment 16. Specify and quantify “much better”.
Comment 17. Change to, the EAS were …… than the ethanolic crude extract of AS”
Line 224, “… And on compounds level, 1-4 showed…”
Comment 18. Remove “And on..” instead use, “Isolated compounds 1-4 showed…”
Line 224-225, “1-4 showed good results…”
Comment 19, Change to, “1-4 showed significantly better” …or… “exhibited significantly better anti-oxidative and anti-inflammatory activity compared to their respective positive controls.”
Author Response
Response to Reviewer 3 Comments
Reviewer 3)
Lines 2-5, “Anti-inflammatory and antioxidant activities of the diarylheptanoid isolated from enzymatic hydrolysed Alnus sibirica by Fungamyl Super AX and comparisons with original plant”
Comment 1. All data and discussion regarding the variation in the content of the isolated diarylheptanoid due to processing using enzymatic hydrolysis must be deleted from this manuscript since this topic has been previously published:
“Wang, H.S., Yin, J., Hwang, I.H., and Lee, M.W., Variation of diarylheptanoid from alnus sibirica fitch. ex turcz. processed enzymatic hydrolysis. Korean Journal of Pharmacognosy, 2018. 49(4): pp. 336-340.”
Inclusion of this data and discussion in this manuscript entails plagiarism.
Furthermore, it is contentious why the authors have not sited this previously published article?
Answer) → Previously published paper ““Wang, H.S., Yin, J., Hwang, I.H., and Lee, M.W., Variation of diarylheptanoid from alnus sibirica fitch. ex turcz. processed enzymatic hydrolysis. Korean Journal of Pharmacognosy, 2018. 49(4): pp. 336-340.” was dealing with only contents analysis of EAS and validation of compounds. And we want to emphasize the differences of fermentation and enzymatic hydrolysis of Alnus sibirica. But we decide to delete the part of “HPLC analysis” from the manuscript.
Comment 2. Change the title to, “Anti-inflammatory and antioxidant activities of the diarylheptanoid isolated from enzymatically hydrolysed ethanolic extract of Alnus sibirica”
Answer) → Modifications completed. “Inhibitory effects on NO production and DPPH radicals and NBT superoxide activities of the diarylheptanoid isolated from Enzymatically hydrolysed ehthanolic extract of Alnus sibirica”
Line 40, “ In our study, 17 compounds from …..”
Comment 3. These were not part of the current manuscript but rather from the previous reported study, this is confusing to readers and should be changed to … “ From our previous study…”
Answer) → Modifications completed :
[FromIn our previous study]
Line 43, “ on EAS including the compositional analysis of the diarylheptanoid…”
Comment 4. Again, remove this part from this manuscript, this has been reported previously (see comment 1).
Answer) → We deleted the content “including the compositional analysis of the diarylheptanoid”
[Present paper describes the evaluation of antioxidant and anti-inflammatory effects on EAS including the compositional analysis of the diarylheptanoid isolated therefrom.]
Line 51-53, “Figure 1. Stuctures of compounds 1-4”
Comment 5. Remove this figure, it has very strong similarity with the previously published article (see comment 1). It is rather recommended to show the hydrolysis reaction scheme with reaction conditions indicated.
Answer) → We revised the Figure 1. that included the hydrolysis reaction scheme.
Figure 1. Sructures of compounds 1-4Chemical structure of 1-5 and hydrolysis reaction pathway
Lines 56. 65, 73, 81, “….1, 2, 3, 4…”
Comment 6. Write instead as, Compound 1, Compound 2 …..”
Answer) → Modifications completed :
[1, 2. 3, ,4 → Compound 1, Compound 2, Compound 3, Compound 4…]
Line 84, “…[d 6.68 (1H, d, J = 7.2, H-5′)…..”
Comment 7. Be consistent in describing coupling constants, always indicate “Hz” after values.
Answer) → Modifications completed :
[d 6.68 (1H, d, J = 7.2, H-5′) → d 6.68 (1H, d, J = 7.2 Hz, H-5′)]
Line 84, “The 1H-NMR spectrum….”
Comment 8. Similarly to all compounds described, indicate the Hz of NMR machine and d-solvents used.
Answer) → Modifications completed :
[The 1H-NMR spectrum… → : The 1H-NMR (600 MHz, DMSO-d6+D2O) spectrum…]
Comment 9. Furthermore, a complete table of 1H-NMR and 13C-NMR values with pertinent NMR information indicated should be presented as table or supplemental texts.
Answer) → Added “Supplementary Material” files that included 1H and 13C-NMR information.
Line 162, “….under vacuum, the mix was…”
Comment 10. Change to, “…., the mixture was …”
Answer) → Modifications completed :
[After removing prethanol A under vacuum, the mix was… → After removing ethanol, the mixture was…]
Line 164, “ … fractionation by ethyl acetate; …”
Comment 11. Be specific on how the fraction was obtained, if it was liquid-liquid partition, then it should be changed to, “… followed by liquid-liquid partition using ethyl acetate and…. The ethyl acetate layer was …..and the …. was stored in the freezer.”
Answer) → Modifications completed :
[…and fractionation by ethyl acetate… → …A part (23.27 g) of this extract was subjected to enzymatic hydrolysis with Fungamyl (to obtain EAS) and followed by liquid-liquid partition using by ethyl acetate. The rest of the extract was stored in the freezer and ethyl acetate layer was then subjected to Sephadex LH-20 column chromatography…]
Line 164-165, “EAS was then subjected to Sephadex LH-20….”
Comment 12. Specify which EAS solution was used in this section, the aqueous, ethanolic or the ethyl acetate layer?
Answer) → The ethyl acetate layer. And modified that part. :
[the rest of the extract was stored in a freezer. EAS was then subjected to Sephadex LH-20 column chromatography… → The rest of the extract was stored in the freezer and ethyl acetate layer was then subjected to Sephadex LH-20 column chromatography…]
Line 167-168, “….was subjected to MPLC ….”
Comment 13. Specify the MPLC conditions (detection wavelength, injection volume etc…) and columns (brand, particle size, i.d.) used.
Answer) → Since ODS gel doesn’t elute unless pressure is applied, MPLC was used only for pressure application. And another MPLC and column information described in “Materials and Methods” as below.
[ ODS-B gel (40–60 μm, Daiso, Japan) was used as the stationary phase on a medium pressure liquid chromatography (MPLC) system and consisted of an injector (Waters 650E), pump (TBP5002, Tauto Biotech, China), and a detector (110 UV/VIS detector, Gilson, Middleton, WI, USA).]
Line 176, “…standing for 30 min…”
Comment 14. Specify incubation parameters (e.g. temperature and light conditions) and detector wavelength used.
Answer) → Modifications completed :
[…standing for 30 min… → … standing for 30 min at 37 °C in dark…]
Line 184, “… a blank reaction was carried out…”
Comment 15. Specify the composition of the blank reactants.
Answer) → Modifications completed :
[… a blank reaction was carried out… → …a blank was carried out by input distilled water instead extract and xanthine oxidase.]
Line 224, “… the EAS were much better than those of AS.”
Comment 16. Specify and quantify “much better”.
Answer) → That was described in “Results and Discussions” part as below.
[ “EAS (IC50 = 16.68 ± 0.37 μg/mL) showed better DPPH radical scavenging activity than AS (IC50 = 21.80 ± 0.55 μg/mL).”
“EAS (IC50 = 3.12 ± 0.75 μg/mL) showed more potent NBT superoxide scavenging activity than AS (IC50 = 4.59 ± 0.68 μg/mL)”
“EAS (IC50 = 1.14 ± 0.06 μg/mL) showed better inhibitory effects on NO production than AS (IC50 = 6.26 ± 0.16 μg/mL)” ]
Comment 17. Change to, the EAS were …… than the ethanolic crude extract of AS”
Answer) → Modifications completed :
[…than those of AS. → … than the ethanolic crude extract of AS.]
Line 224, “… And on compounds level, 1-4 showed…”
Comment 18. Remove “And on..” instead use, “Isolated compounds 1-4 showed…”
Answer) → Modifications completed :
And on compounds level, 1-4 showed…→ Isolated compounds 1-4 showed…
Line 224-225, “1-4 showed good results…”
Comment 19, Change to, “1-4 showed significantly better” …or… “exhibited significantly better anti-oxidative and anti-inflammatory activity compared to their respective positive controls.”
Answer) → Modifications completed :
[1-4 showed good anti-oxidative… → 1-4 showed significantly better anti-oxidative and anti-inflammatory activity compared to their respective positive controls.]

Round 2
Reviewer 1 Report
None
Reviewer 2 Report
The authors responded to almost all the recommendations.
“Chemical section” refers to the sources of the substances and solvents used for the analysis.
There are still a lot of language mistakes
The manuscript can be still improved.